# An arginine switch drives the stepwise activation of β-arrestin by CXCR7

Jeong Seok Ji[1]☉, Yaejin Yun[1]☉, Tomasz Maciej Stepniewski[2,3,4]☉, Hye-Jin Yoon[1], Kyungjin Min[1], Ji Young Park[5], Chiwoon Chung[5], Miguel Dieguez Eceolaza[2], Ka Young Chung[5], Jana Selent[2]*, Hyung Ho Lee ᴵᴰ[1]*

1 Department of Chemistry, College of Natural Sciences, Seoul National University, Seoul, Republic of Korea, 2 Research Program on Biomedical Informatics, Hospital del Mar Research Institute and Pompeu Fabra University, Barcelona, Spain, 3 Faculty of Chemistry, Biological and Chemical Research Centre, University of Warsaw, Warsaw, Poland, 4 InterAx Biotech AG, Villigen, Switzerland, 5 School of Pharmacy, Sungkyunkwan University, Suwon, Republic of Korea

☉ These authors contributed equally to this work.
* jana.selent@upf.edu (JS); hyungholee@snu.ac.kr (HHL)

## Abstract

β-arrestins (βarrs) play a crucial role in regulating G protein-coupled receptor (GPCR) signaling and trafficking. Canonically, interactions of βarr with the phosphorylated intracellular GPCR-tail induce a multi-step conformational transition that results in the activation of βarr. Depending on the specific interaction pattern with the receptor, βarrs adopt multiple conformational states, each tightly linked to a specific functional outcome of βarr recruitment. Despite its physiological relevance, the structural determinants of βarr activation remain poorly understood. Using a combination of molecular dynamics simulations, biochemical and cell-based experiments, we reveal how specific interactions with a chemokine receptor 7 (CXCR7) promote the unbinding of the βarr2 C-tail—a crucial step in arrestin activation. Importantly, we observe that the expulsion of the C-tail is promoted by the displacement of a conserved arginine residue (Arg394) within the βarr polar core, which we dub "the arginine switch." Our study uncovers a role for the arginine switch that, upon engagement, destabilizes the polar core as a crucial step in the CXCR7-induced βarr activation.

## Introduction

β-arrestin1 and β-arrestin2 (referred to as βarr1 and βarr2) are multifunctional adaptor proteins that mediate the desensitization and internalization of G protein-coupled receptors (GPCRs) [1–4]. Upon agonist-induced activation of GPCRs, βarrs bind to the phosphorylated C-tails or intracellular loops of the receptor [5–10]. This leads to the recruitment of clathrin and the adaptor protein complex-2 (AP2), which triggers receptor endocytosis [11]. Subsequently, the internalized receptor can be either recycled back to the cell surface or degraded [11–13]. The interaction between βarrs and clathrin/AP2 complexes is crucial for the efficient internalization of GPCRs [1,14]. The removal of

**Data availability statement:** The MD simulation data will be publicly available as of the date of publication at the GPCRmd repository (www.gpcrmd.org) (https://www.gpcrmd.org/dynadb/publications/1525/).

**Funding:** H.H.L. was supported by grants from the National Research Foundation (NRF) of Korea funded by the Korean government (2022R1A2B5B02002529 and 2022R1A5A6000760) and by the Bio & Medical Technology Development Program of the National Research Foundation (NRF) funded by the Korean government (MSIT) (RS-2024-00396026). T.M.S. was supported by resources from Sara Borrell grant CD22/00007 funded by the Institute of Health Carlos III (ISCIII), resources of grant 2021 SGR 00046 funded by Agència de Gestió d'Ajuts Universitaris i de Recerca Generalitat de Catalunya (AGAUR), and the National Center of Science, Poland, grant 2017/27/N/NZ2/02571. J.S. acknowledges funding from MICIU/AEI/10.13039/501100011033 and the ERDF/EU (grant number PID2022-137161OB-I00). J.S. acknowledges further funding from the Horizon Europe Project OBELISK under the grant agreement 101080465. K.Y.C. was supported by NRF of Korea (2021R1A2C3003518). The funders had no role in study design, data collection and analysis, decision to publish, or preparation of the manuscript.

**Competing interests:** The authors have declared that no competing interests exist.

**Abbreviations:** 3E, three hydrophobic elements; βarr, β-arrestin; AP2, adaptor protein complex-2; CXCR7, chemokine receptor7; GPCR, G protein-coupled receptor; HDX-MS, hydrogen/deuterium exchange mass spectrometry; ITC, isothermal titration calorimetry.

GPCRs from the cell surface through βarr-mediated internalization limits the continued transmission of extracellular signals into the cell. However, despite being initially identified as proteins that desensitize GPCRs, βarrs can also interact with other signaling proteins, including mitogen-activated protein kinases [15–16], Src protein kinase [17], AKT [18], and the NF-κB cascade [19–21], triggering a second wave of intracellular signaling, independent of G proteins [22–24]. The development of 'biased' drugs, which selectively modulate the activation of GPCRs to preferentially stimulate or avoid stimulating βarrs instead of G proteins, has the potential to yield improved therapeutic outcomes with enhanced safety profiles for a broad spectrum of diseases [25].

A hallmark of β-arrestin (βarr) is its remarkable structural flexibility. In response to interactions with GPCRs and small molecules, the protein can undergo a conformational transition in a process called activation [26]. βarr can assume multiple active conformations, which are linked to how it engages with the receptor and influence the functional outcome of βarr recruitment [27–30]. Therefore, it is crucial to understand the molecular mechanisms governing the activation of βarrs by GPCRs.

Previous structural studies have identified both inactive and active states of βarrs. The inactive state is stabilized through the autoinhibitory binding of the βarr C-terminal tail to the N-domain of βarrs [24,31–33], which prevents the inter-domain rotation between the N- and C-domains. The autoinhibitory binding of βarr C-tail to the N-domain occludes the engagement of the receptor C-tail (hereafter referred to as the $R_p$-tail). Thus, it has been hypothesized that the binding of $R_p$-tail or intracellular loops of GPCRs to the N-domain of βarrs is associated with the release of βarr C-tail from the N-domain [24,34–36]. Indeed, the release of βarr C-tail from the N-domain and the subsequent inter-domain rotation have been observed in the active state of βarrs [8–10,26,37–42].

The autoinhibitory binding of βarr C-tail to the N-domain is sustained by two distinct interfaces: the polar core and three hydrophobic elements (3E) (β-strand I and α-helix I in the N-domain and β-strand XX in the C-tail) [32,33,37,43]. The release of the βarr C-tail is a critical step in facilitating downstream signaling by exposing buried regions that are otherwise inaccessible in inactive βarr [32,44,45]. These regions include the βarr C-tail regions responsible for clathrin binding, as well as the N-domain regions of βarr that bind to scaffolding signaling kinases [14,24,46,47]. Despite the functional significance of the release of the βarr C-tail, the molecular mechanism underlying how $R_p$-tail binding promotes the full displacement of the βarr C-tail remains unclear due to the absence of structural information.

In this study, we aimed to unravel initial states of βarr2 activation by a $R_p$-tail. To achieve this, we simulated βarr2 in complex with a phosphopeptide (C7pp2) derived from the carboxyl terminus of CXCR7, an atypical chemokine receptor that interacts with βarrs but lacks functional coupling with heterotrimeric G proteins [48]. Our structural analysis suggests that C7pp2 can bind while the βarr2 C-tail remains attached to the N-domain of βarr2, with a partially disrupted polar core and other characteristics resembling its inactive state. These structural insights, together with biochemical and cell biological experiments, reveal the arginine switch mechanism and shed light on the activation mechanism of βarrs by CXCR7 C-tail.

## Results

### An intermediate state of βarr2

A previous study assessed the functional significance of CXCR7 phosphorylation in βarr trafficking and elucidated that a distal phospho-site cluster, PxPxxP (C7pp2, where P represents pSer/pThr and x is any other amino acid, Fig 1A), significantly contributes to βarr2 recruitment and trafficking [56]. Since βarr2 is known to be stable in an auto-inhibited basal state, where its C-tail containing β-strand XX is bound to the N-domain, we hypothesized that this auto-inhibited state could only be destabilized by sufficient phosphorylation at the GPCR $R_p$-tail. To verify whether the presence of 3 phosphates (pSer350, pThr352, and pSer355) in C7pp2 results in the release of βarr2 C-tail, we performed an *in vitro* clathrin binding assay using full-length βarr2 [57]. The clathrin binding motif of βarr2 (residues 371-379) is only accessible for clathrin binding when βarr2 C-tail is released. Our results showed that C7pp2 binding to βarr2 does not enhance clathrin binding compared to $V_2$Rpp, a well-characterized phosphopeptide derived from the C-terminus of the vasopressin type 2 receptor [57], which effectively leads to the release of βarr2 C-tail (Fig 1B). In contrast, in the case of C7pp3, which harbors additional phosphorylations (pSer347, pSer360, and pThr361), the βarr2 C-tail is released (Fig 1B). These results indicate that the phosphorylation of C7pp2 at three sites is insufficient to fully displace the βarr2 C-tail.

To better understand the interaction dynamics where the βarr2 C-tail is not fully displaced when bound to C7pp2, we used a sequential modeling protocol of molecular dynamics simulations. First, we simulated the re-binding of the C-tail, starting from an unbound conformation, to explore intermediate conformational states of the C-tail (35 × 200 ns, Fig 1C). Our analysis specifically focused on Arg394 within the distal C-tail, as previous experimental findings have underscored the conformational rearrangement of this residue as an important step in C-tail displacement [57]. In the inactive conformation of βarr2, Arg394 is stabilized by interactions with Asp27 and Asp299 (Fig 1C). Monitoring the distance between Arg394 and Asp299 revealed that, in the majority of the simulation replicates, we observed spontaneous inactivation of the partially displaced distal C-tail (S1 Fig and S1 Movie). Interestingly, when clustering the conformational landscape explored by Arg394 during these transition dynamics (Fig 1D), we observed a highly populated cluster (cluster I, 50%) of conformations where Arg394 is extended away from the βarr2 surface. The relatively high population of this conformation suggests that it represents an important intermediate state explored by the distal C-tail during binding/unbinding. To verify whether such a conformation can accommodate interactions with an $R_p$-tail, we docked the phosphorylated C7pp2 to the representative βarr structure of cluster I. Intriguingly, the docking algorithm generated a pose in which Glu358 of C7pp2 forms interactions with the partially displaced Arg394 of βarr2 (Fig 1E). Thus, our structural model suggests that Glu358 of C7pp2 stabilizes the displaced Arg394 of βarr2 and, in doing so, supports the initial stage of distal C-tail displacement in an intermediate active state of βarr2 (βarr2$_{IM}$).

To further study the obtained βarr2$_{IM}$-C7pp2 complex, we modeled the entire distal C-tail, which is typically not resolved in experimental structures due to its high flexibility, and carried out MD simulations for structural relaxation (Fig 1F). Intriguingly, clustering analysis of the resulting simulation runs revealed that Arg394 of βarr2 primarily adopts conformations where it maintains electrostatic interactions with Glu358 of C7pp2 (Fig 1F).

### Two distinct binding motifs between βarr2$_{IM}$ and C7pp2

A more detailed analysis of the interface between C7pp2 and βarr2$_{IM}$ reveals two attachment points where C7pp2 forms polar interactions with positive residues on the βarr2$_{IM}$ surface, referred to as site I and site II (Fig 2A). To gain insights into the conformational flexibility of the generated complex, we conducted unrestrained MD simulations with five replicates, each running for 300 ns (S3 Fig). When analyzing individual snapshots over the simulation time, we found that the distal C-tail (blue), despite being highly flexible, stacks on top of C7pp2, thereby maintaining frequent interactions with it (S4 Fig). Additionally, we observed differences in the stability of the main contact sites I and II of C7pp2. Residues within site II explored a wider range of conformations compared to site I (S4 Fig). This variability is also reflected in the frequency of

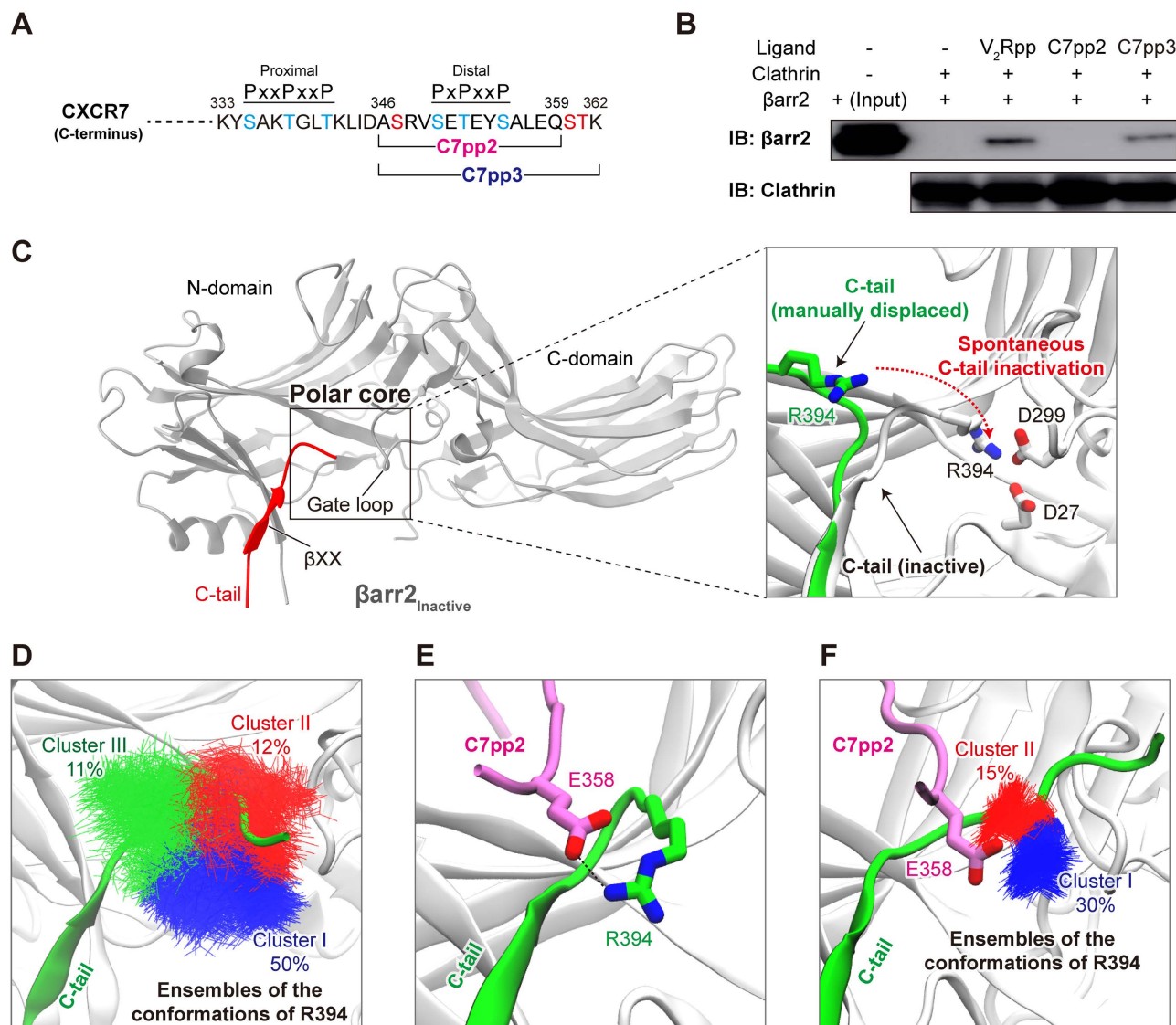

**Fig 1. Generating a model of the intermediate state of βarr2. (A)** The C-terminal sequence (residues 333-362) of CXCR7. The C7pp2 (residues 346-359) and C7pp3 (residues 346-362) peptides are indicated. Phosphorylation sites in the proximal and distal clusters (PxxPxxP and PxPxxP, where P represents pSer/pThr and x represents any other amino acid) are indicated in blue, while added phosphorylation sites on C7pp3 are denoted in red. All phosphorylated residues have been experimentally validated [49–55]. **(B)** *In vitro* clathrin binding assay for βarr2. The βarr2 binding and input are shown. Clathrin binding to βarr2 was enhanced upon binding of V$_2$Rpp and C7pp3. **(C)** Ribbon diagram of inactive βarr2. The C-tail of inactive βarr2 is highlighted in red. The polar core region, gate loop, and βXX motif are indicated. A zoomed-in view of the polar core region (right panel) shows an initial step of the simulation. The C-tail of βarr2 (residues 385–397) was manually displaced (shown in green) from the initial inactive conformation. The observed spontaneous inactivation of the C-tail during the simulation is indicated by a dotted arrow. **(D)** Representative conformations of R394 during spontaneous inactivation of the βarr2 C-tail. The frames obtained from simulations of the generated model (35×200 ns, restraints applied to the backbone of βarr2) were clustered based on the conformation of R394. The ensembles of the hit conformations of R394 are shown in blue (cluster I), red (cluster II), and green (cluster III), respectively. **(E)** The βarr2-C7pp2 complex model obtained from docking analysis. The most populated conformation of βarr2 obtained from clustering analysis was used to dock the C7pp2 peptide. **(F)** The representative conformations of R394 in each cluster from the simulation results. The ensembles of the hit conformations of R394 are shown in blue (cluster I) and red (cluster II). The βarr2-C7pp2 complex model obtained from docking result was used for simulations (35×300 ns, last 50 ns taken for analysis).

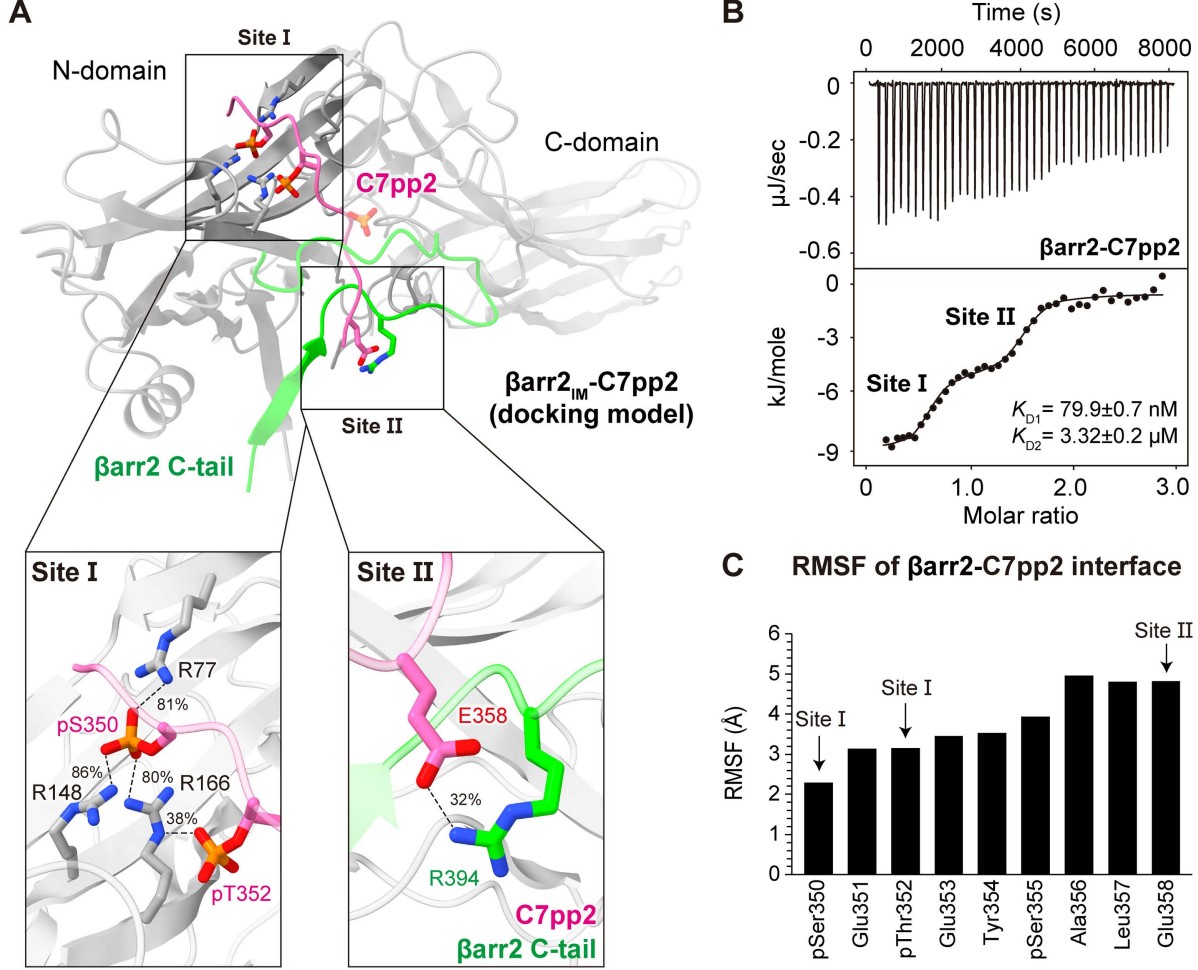

**Fig 2. Model of the intermediate βarr2-C7pp2 state. (A)** Structural model of the intermediate βarr2-C7pp2 complex. Interactions between C7pp2 and clusters of positive residues in βarr2 have been categorized into site I and site II, respectively. **(B)** ITC analysis of the binding of C7pp2 to βarr2. Purified βarr2 proteins were incubated with increasing concentrations of C7pp2 peptide, and the binding parameters were calculated based on the dose-response curve. βarr2 induces biphasic binding by C7pp2. The indicated $K_{D1}$ and $K_{D2}$ values correspond to site I and site II, respectively. **(C)** RMSF analysis of the Cα atoms of the simulated C7pp2 peptide. Residues of C7pp2 belonging to sites I or II are indicated with arrows. The data underlying the graphs shown in the figure can be found in S1 Data.

contacts established between C7pp2 and βarr2$_{IM}$. For site I, we found very stable electrostatic interactions between the phosphorylated Ser350 in C7pp2 and Arg77 (81%), Arg148 (86%), and Arg166 (80%) in βarr2$_{IM}$ (Fig 2A). Furthermore, the phosphorylated Thr352 in C7pp2 contributes to the stability of site I by interacting with Arg166 (38%) of βarr2$_{IM}$. Interestingly, for site II, our simulations revealed transient interactions between Glu358 of C7pp2 and Arg394 of βarr2$_{IM}$, which can break and reform multiple times during a single simulation run, achieving a total stability of 32% within the simulation frames (Fig 2A). Despite their highly dynamic nature, the observed interactions appear to primarily converge over the course of the simulations in the studied replicates (S5 Fig).

Notably, the structural model is supported by isothermal titration calorimetry (ITC) experiments (Fig 2B). In these experiments, βarr2 exhibits a biphasic binding profile with C7pp2, and fitting the data to a two-site binding model suggests that one or two C7pp2 peptides can bind to βarr2. This non-physiological 1:2 stoichiometry (receptor:βarr2 = 1:1 in cells) likely arises from the use of the isolated C-terminal peptide of CXCR7 rather than the full-length receptor. Nonetheless,

the biphasic behavior clearly indicates that C7pp2 harbors two distinct binding motifs for βarr2—one that interacts at lower concentrations and another that becomes engaged only at higher concentrations during the titration. In this context, our structural model (Fig 2A) suggests that the primary ITC binding site (with higher binding affinity) corresponds to site I in our βarr2$_{IM}$-C7pp2 complex, which features abundant electrostatic interactions, whereas the second ITC binding site (with significantly lower affinity) corresponds to site II. This is further supported by the RMSF profile obtained from MD simulations (Fig 2C), which reveals greater binding stability for site I (lower RMSF values) compared to site II (higher RMSF values).

To provide experimental evidence supporting our structural model, we disrupted the N-terminal and C-terminal interactions of C7pp2 with βarr2$_{IM}$ using R166A (site I) and R394A (site II) mutants. Remarkably, these mutations abolished the binding peaks characteristic of site I and site II, respectively (S6 Fig). Our results indicate that C7pp2 binds to βarr2 through two distinct binding events: first, the high-affinity binding of the C7pp2 N-terminus at site I (with a $K_D$ of 1.1 μM for the R394A mutant), and second, the low-affinity binding of the C7pp2 C-terminus at site II (with a $K_D$ of 16.4 μM for the R166A mutant).

To further validate the contributions of pSer350 and pThr352 in the binding of C7pp2 to βarr2$_{IM}$ at site I of our structural model (Fig 2A), we substituted these residues with non-phosphorylated Ser350 and Thr352 and assessed the effects of each mutation using ITC (S6 Fig). Importantly, we found that the site I binding peak was abolished, confirming the crucial role of pSer350 and pThr352 in site I binding. Taken together, we conclude that C7pp2 binds to βarr2 through two distinct binding interfaces, while the βarr2 C-tail remains attached to the N-domain of βarr2.

## Partially disrupted polar core in βarr2$_{IM}$ by C7pp2

Notably, our structural model suggests the existence of an intermediate state with interactions between Glu358 of C7pp2 and Arg394 of the βarr2 polar core (Fig 3A). Strikingly, this structural arrangement seems to stabilize a partially displaced C-tail conformation. It is widely recognized that the polar core interactions and 3E, namely β-strand I and α-helix I in the N-domain and β-strand XX in the C-tail, play crucial roles in stabilizing the basal state of βarr2 [31,33]. Consequently, both polar core and 3E must be disrupted in an active state of βarr2, which is the current conceptual framework of GPCR-βarr2 interaction. In this respect, our structural model suggest that the polar core can be partially destabilized, while the 3E interaction remains intact (Fig 3A).

The polar core of βarr2 in the basal state (Fig 3A) is comprised of charged residues from N-domain (Asp27 and Arg170), C-domain (Asp292 and Asp299), and C-terminal tail (Arg394), bringing different parts of βarr2 together, i.e., the N-domain, C-domain, and C-terminal tail hold together to prevent inter-domain rotation, which is a landmark structural change during βarr2 activation [43]. Among them, Arg394 points to the deep cleft between N- and C-domain, forming a central salt bridge, restrained by Asp27 (N-domain) and Asp299 (gate loop). It is interesting to note that Arg394 is the only residue that directly bridges all three parts (N-domain, C-domain, and C-terminal tail) among the five residues of the polar core.

To computationally verify the stabilizing effect of C7pp2 on the intermediate state, we monitored the integrity of the polar core by measuring the distance of Arg394 to Asp299 of βarr2$_{IM}$ (Fig 3B). While small values indicate an intact polar core (i.e., inactive βarr2) large distances reflect polar core disruption (i.e., intermediate βarr2). Interestingly, we find in the βarr2$_{IM}$-C7pp2 complex that the polar core remains largely disrupted as indicated by an average large distance of 10 Å (Fig 3B). In contrast, when removing the C7pp2, we can observe the re-formation of a salt bridge between Arg394 to Asp299 of βarr2 in several replicates (Fig 3B, average distance 6 Å). This suggests that the binding of C7pp2, specifically the interaction of Glu358 in C7pp2 with Arg394 is a critical element to stabilize a broken polar core and in turn an intermediate βarr2 state.

To further corroborate this finding, we performed an *in silico* mutation of Glu358 to alanine (E358A) in the C7pp2. Remarkably, we found that this mutant also shows βarr2 inactivation in several replicates starting from a partially

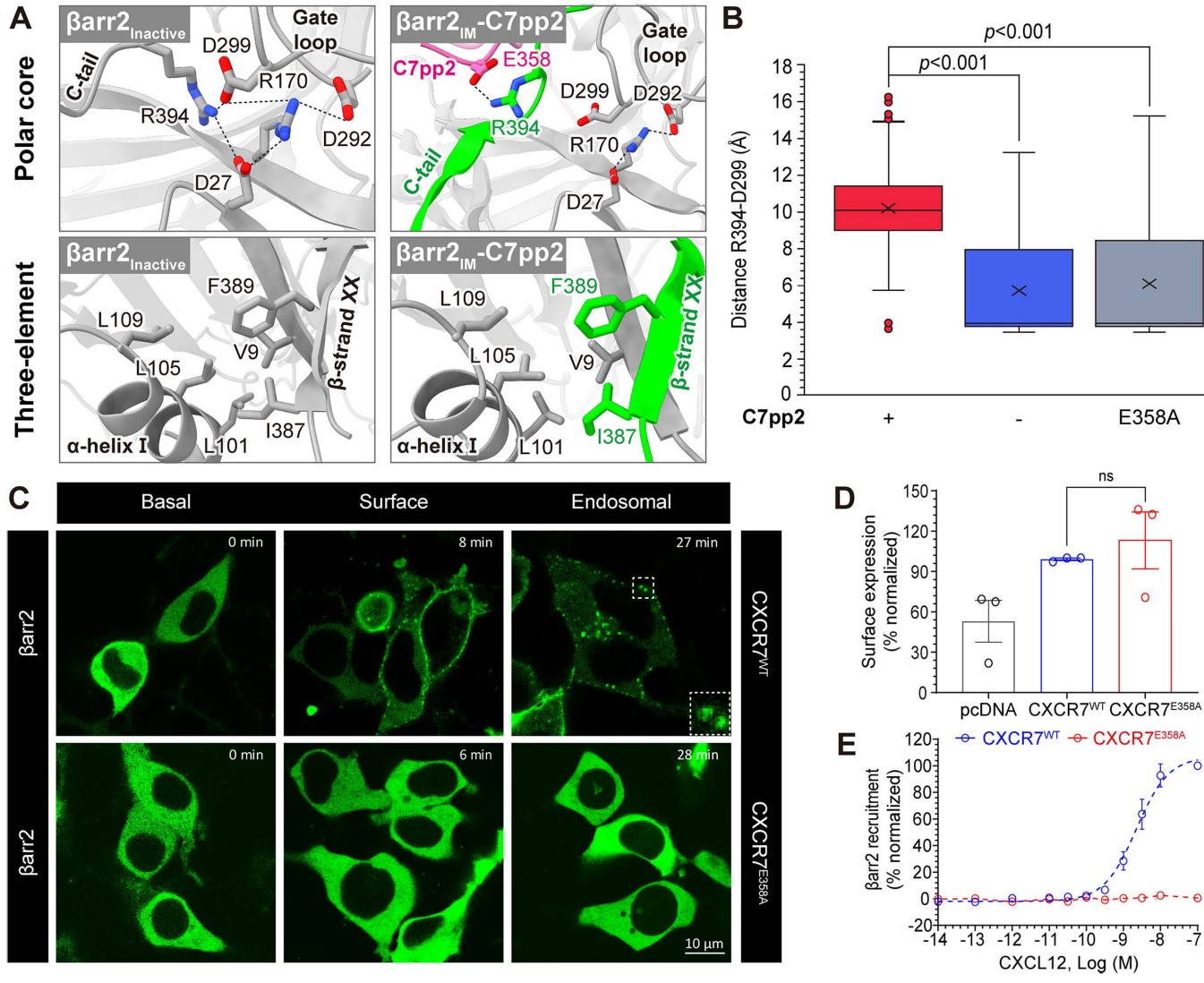

**Fig 3. Destabilization of the polar core by C7pp2. (A)** Comparison of polar core and three-element interactions between the inactive and intermediate conformations of βarr2. The ribbon diagram of βarr2 is colored dark gray. C7pp2 and the C-tail of βarr2_IM are shown in pink and green, respectively. Hydrogen bonds stabilizing the polar core interactions are shown as dashed lines. **(B)** Boxplots comparing the distance between D299 and R394 (forming polar interactions in the inactive βarr2) monitored within MD simulations (5×300 ns). Outliers are depicted as points. The significance of differences between the groups was measured using a two-sided Mann–Whitney $U$ test. **(C)** CXCL12-induced trafficking of βarr2 as monitored using confocal microscopy in HEK293 cells expressing CXCR7^WT and CXCR7^E358A. The "Surface" images depict βarr2 localization at the plasma membrane shortly after agonist stimulation, representing initial recruitment of βarr2 to CXCR7. The "Endosomal" images show intracellular βarr2 localization at later time points, indicating receptor internalization. Scale bar is 10 μm. **(D)** The whole-cell ELISA-based assay to measure the surface expression of CXCR7. Data are presented as mean ± SEM of three independent experiments, normalized to CXCR7^WT. **(E)** CXCL12-induced βarr2 recruitment to CXCR7^WT and CXCR7^E358A in the Tango assay. Data are presented as mean ± SEM. CXCR7^E358A is significantly compromised in inducing βarr2 trafficking upon CXCL12 stimulation. The data underlying the graphs shown in the figure can be found in S1 Data.

displaced C-tail (Fig 3B, average distance 6 Å) emphasizing the role of Glu358 of C7pp2 and Arg394 of βarr2 interactions in stabilizing the intermediate state. To experimentally verify this observation, we conducted ITC experiments, which showed that binding at site II was indeed lost in the C7pp2^E358A mutant as well as in the βarr2^R394A mutant (S6 Fig, lower panel).

Moreover, confocal microscopy revealed that, unlike CXCR7[WT], the CXCR7[E358A] mutant failed to induce βarr2 redistribution from the plasma membrane to intracellular endosomes upon CXCL12 stimulation, suggesting impaired receptor internalization (Fig 3C). Additionally, the Tango assay demonstrated that CXCR7[E358A] exhibited a significant reduction in βarr2 recruitment in response to CXCL12, confirming that the E358A mutation disrupts βarr2 engagement and downstream trafficking (Fig 3E). While the observed loss of function does not definitively confirm the existence of an intermediate state, it does provide indirect support for the role of Glu358 in the activation process of βarr2. Altogether, these results strongly suggest that the destabilization of the polar core is initiated by the formation of a salt bridge between Glu358 of C7pp2 and Arg394 of βarr2$_{IM}$.

## HDX profile changes of βarr2 upon C7pp2 binding

To further study the conformational dynamics of βarr2 upon C7pp2 binding in solution, we adopted hydrogen/deuterium exchange mass spectrometry (HDX-MS) (Figs 4 and S7 and S1 Data). HDX-MS monitors the exchange between hydrogen atoms in the peptide bond backbone and deuterium in the solvent [58]. The exchange rate is affected by the conformational dynamics (i.e., exposure to the buffer and conformational flexibility) of the local area of the proteins [58]. Upon co-incubation with C7pp2, βarr2 showed increased HDX at the gate loop (Fig 4, peptides 292–302), which is likely due to the disruption of the interaction between the gate loop (Asp299) and C-tail (Arg394). Unfortunately, we could not obtain HDX-MS data from the distal C-tail region so that we could not investigate its conformational changes. However, the HDX profile of N-terminal part of βXX was increased (Fig 4, peptides 382–389) potentially due to increased conformational dynamics allosterically transmitted from the breakage between Asp299 and Arg394. This result supports our hypothesis that the disruption of the interaction between the gate loop (Asp299) and C-tail (Arg394) by C7pp2 further destabilizes β-strand XX. The HDX rates were decreased in the βVI through middle loop, lariat loop, and back loop (Fig 4, 119–133, 283–291, and 305–317). These observations raise the possibility that binding of C7pp2 to βarr2 could induce conformational rearrangements throughout the βarr2 domains. Overall, our results suggest that C7pp2 binding affects primarily

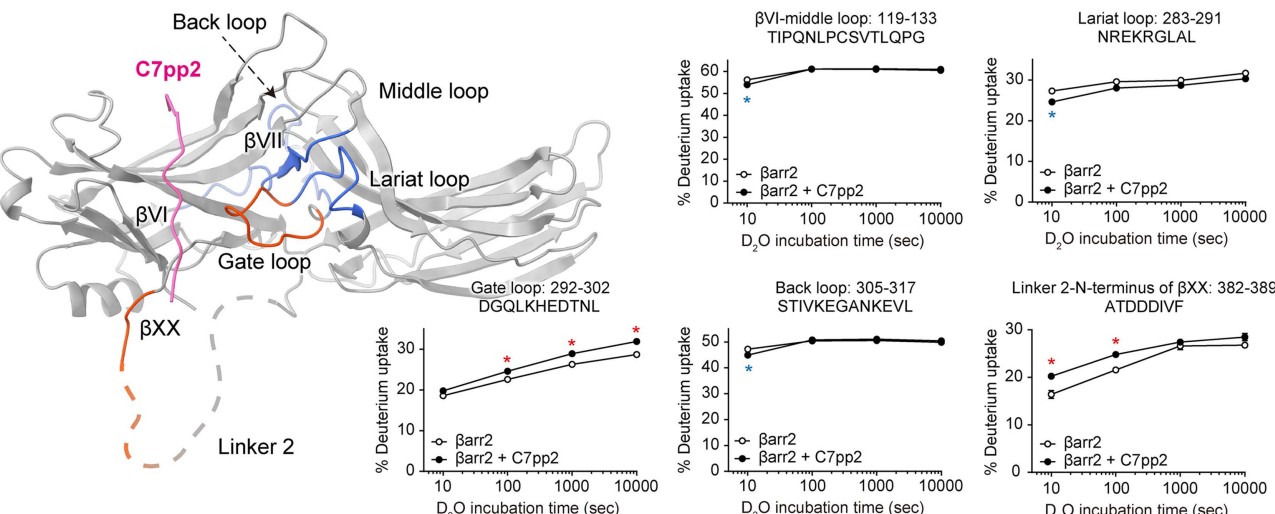

**Fig 4. HDX profile changes of βarr2 upon co-incubation with C7pp2.** Regions with increased or decreased HDX upon co-incubation with C7pp2 are colored red or blue, respectively, on the structure of the βarr2$_{IM}$-C7pp2 complex and the HDX profiles of the corresponding peptides are shown as graphs. Data represent the mean ± standard error of three independent experiments. Statistical analysis was performed using Student *t* test (*$p < 0.05$ compared with βarr2 alone). Differences smaller than 0.2 Da were not considered significant. The data underlying the graphs shown in the figure can be found in S1 Data.

parts of the βarr2 C-tail through its partial displacement as well as the dynamics of the gate loop through polar core disruption.

## Discussion

To activate βarr by GPCRs, interactions between GPCRs and βarr occur through two distinct interfaces. The first interface involves the binding of the receptor's $R_p$-tail to the N-domain of βarr. The second interface involves the interaction between the receptor's transmembrane helices and cytoplasmic loops, also known as the receptor core, with the central crest loops of βarr [24,34,35,59]. In this regard, there are at least three distinct binding modes between GPCRs and βarrs, which can be mediated by both the receptor core and $R_p$-tail, $R_p$-tail only, or the receptor core only. It has been demonstrated that the receptor core and $R_p$-tail can independently stimulate βarr activation, and these interactions trigger the release of the sequestered βarr C-tail and inter-domain rotation between the N- and C-domains [59]. One of the missing pieces of the βarr activation mechanism is how the sequestered βarr C-tail is released via interaction with GPCRs, which is considered to be the rate-limiting step [60,61].

The previously determined structures of βarr in complex with $R_p$-tails were limited to the inactive and active states of βarr. Consequently, it was not possible to directly observe how the polar core can be destabilized, as it was either in a stable or disrupted state in those structures. In this study, we aimed to investigate the process of initial βarr activation upon binding to the CXCR7 $R_p$-tail with MD simulations. The model of the intermediate state revealed several novel features that differentiate it from existing βarr structures bound to various $R_p$-tails, providing valuable structural insights into the activation mechanism of βarr by the receptor CXCR7 $R_p$-tail.

Firstly, we observed a direct binding between Glu358 of C7pp2 and Arg394 of the βarr2 C-tail. This interaction pulls the βarr C-tail away from the polar core resulting in partial disruption of the polar core through the Arg394 displacement.

Secondly, we found that C7pp2 binds to βarr2 while its C-tail remains bound to the N-domain in the proposed intermediate βarr2-C7pp2 complex. This differs from the canonical binding mode of $R_p$-tails, which involves the displacement of the βarr2 C-tail. As a result, we observed two distinct binding motifs between βarr2 and C7pp2, namely the canonical binding site I and the newly discovered site II, which includes the Arg394 interaction (Fig 2A). This observation is significant as it captures an intermediate activation state that cannot be observed in the existing active βarr2 structures. Consistently, the ITC results support the conclusion that the binding affinity of site I is significantly higher than that of site II, indicating that the binding of C7pp2 will occur sequentially from site I to site II (Fig 2B).

Thirdly, despite the partial disruption of the polar core, the 3E interaction is still maintained. This may be attributed to the absence of two additional phosphorylations at Ser360 and Thr361, which follow the distal phospho-site cluster PxPxxP of C7pp2 (Fig 1A). If phosphorylated Ser360 and Thr361 residues were present at the C-terminus of C7pp2, they would reside in close proximity to the 3E interaction site and disrupt it, as observed in many other active state βarr structures [37,62,63]. Indeed, $V_2$Rpp, which possesses sufficient phosphorylation sites at its C-terminus, efficiently breaks the 3E interaction and triggers the release of βarr C-tail (Fig 1B). Consistently, the clathrin binding assay confirmed that C7pp3, which contains phosphorylations on Ser360 and Thr361 similar to $V_2$Rpp (Fig 1A), enables the release of βarr C-tail (Fig 1B). The phosphorylations on Ser360 and Thr361 appear to be mediated by GRK2 and may have an independent role, as they differ from the proximal site phosphorylated by GRK5 or the distal site phosphorylated by both GRK2 and GRK5 [63].

Based on these results, we propose the arginine switch model, in which Arg394 binds to C7pp2 to induce βarr activation (Fig 5). In the inactive state, the arginine switch is part of the polar core, maintaining the interaction, and the βarr C-tail interacts with the N-domain through the 3E interaction. When CXCR7 $R_p$-tail binds to βarr through sites I and II, the arginine switch rotates and directly binds to Glu358 of CXCR7 $R_p$-tail, partially disrupting the polar core. This step represents an intermediate state before βarr becomes fully active, and the 3E interaction is still maintained. The phosphorylations of Ser360/Thr361 contribute to breaking the 3E interaction, leading to the release of the βarr C-tail, resulting in βarr folding into a fully active state.

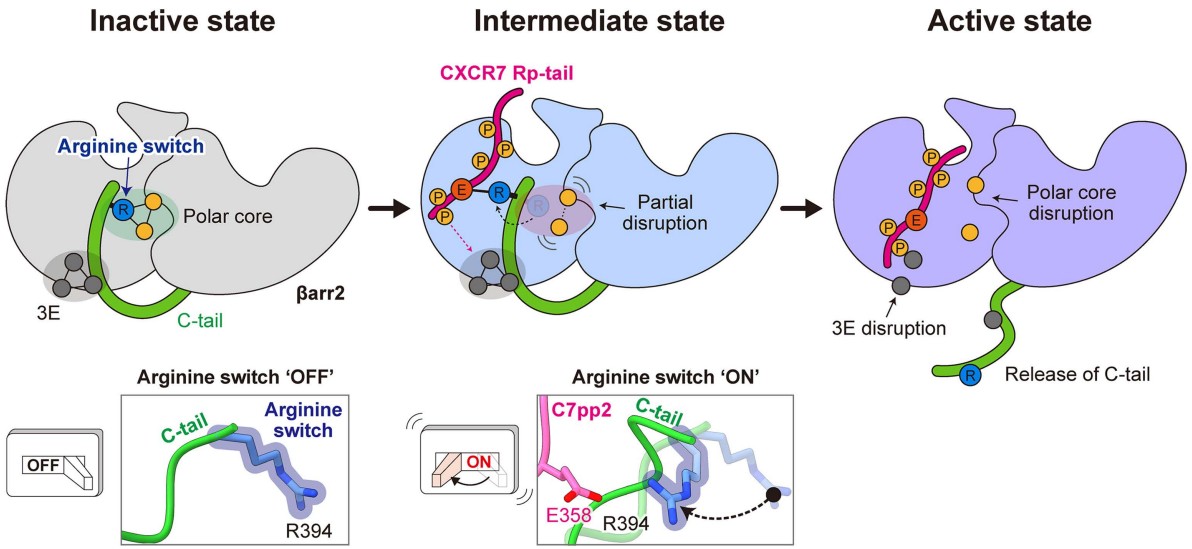

**Fig 5. Proposed arginine switch model for βarr2 activation by the CXCR7 Rp-tail.** In the inactive state, Arg394, which we name as an "arginine switch," is an integral component of the polar core, facilitating interaction. Simultaneously, the βarr2 C-tail engages with the N-domain through a 3E interaction. Upon C7pp2 binding to βarr2, the arginine switch undergoes rotation and forms a direct interaction with Glu358 of C7pp2, leading to partial disruption of the polar core. This stage represents an intermediate state preceding βarr2 activation, where the 3E interaction remains intact. Phosphorylation of Ser360/Thr361 further contributes to the disruption of the 3E interaction, triggering the release of the βarr2 C-tail, ultimately leading to the activation of βarr2.

While our experiments elucidate the activation mechanism of βarr2 by the CXCR7 $R_p$-tail, further investigations will be necessary to fully understand the activation mechanism of βarr. Firstly, this study focuses on CXCR7, and whether the proposed activation mechanism applies to other GPCRs remains to be determined. Although its C-terminal tail shares similarities with other receptors, broader structural comparisons and validation using diverse GPCR phosphopeptides and MD simulations are needed to evaluate the mechanism's general relevance. Secondly, although we explored the activation mechanism of βarr by the receptor $R_p$-tail, our study did not investigate the contributions of the core region of GPCR to βarr activation [7,59,64,65], thus requiring further structural studies of βarr in complex with full-length GPCR. We hypothesize that the GPCR core might also contribute to accelerating the destabilization of the polar core and 3E interactions. Alternatively, the binding of the GPCR core to βarr might trigger additional conformational changes in βarr. It is also important to underline that, although the displacement of Arg394 represents a crucial transition state in the displacement of the βarr C-tail, we cannot exclude the existence of additional transient states prior to the complete unbinding of the C-tail [66]. Thirdly, the possibility of βarr activation by phosphorylated intracellular loops of GPCRs should be considered. In the case of D2-like dopamine receptors, which have shorter C-tails compared to other GPCRs, it has been shown that phosphorylated intracellular loops can interact with βarr and trigger its activation [67,68]. Fourthly, it is necessary to investigate the biological roles of the previously characterized proximal phospho-site cluster (C7pp) [40] and the distal phospho-site cluster (C7pp2). In the case of PTH1R in living cells, βarr2 has been found to engage with both the distal phospho-site cluster and proximal phospho-site cluster, resulting in two distinct complex forms known as the tail-engaged 'hanging' complex and the 'core' complex, respectively [69]. Similarly, it is conceivable that βarr2 could adopt distinct conformations depending on the phospho-site clusters of CXCR7 while associating with the same GPCR, leading to a diverse range of functional outcomes that necessitate further investigation. Lastly, it is possible that the intermediate state of βarr may have its own functional roles in cells, extending beyond solely serving as an activation intermediate state. In a recent report, we demonstrated that the inactive state of βarr can act as a scaffold protein and mediate its functional role

[70]. Therefore, uncovering the potential biological significance of this complex, if any, awaits further investigation in future studies.

As our understanding of G protein-biased or βarr-biased ligand mechanisms expands, we become increasingly aware of their potential therapeutic benefits [71]. While βarr-biased signaling produces positive effects, G protein-dependent signaling can lead to side effects, as observed with carvedilol for both $\beta_1AR$ and $\beta_2AR$ subtypes [72,73], [Sar[1], D-Ala[8]] angiotensin II (TRV120027) for angiotensin II type 1 receptor [74], and PTH for PTH1 receptor [15]. In this case, βarr-biased ligands offer a promising avenue for the discovery of novel drugs, potentially associated with reduced side effects. Our elucidation of the βarr2-C7pp2 complex provides structural insights into the intermediate state through which βarr is activated, thus laying the foundation for pharmacological studies on βarr-biased signaling.

## Materials and methods

### Molecular dynamics simulation and data analysis

For MD simulation, we have started from the structure of βarr2 in an inactive conformation (PDB: 3P2D) [43]. The structure was curated, including the modeling of missing residues and protonation state assignment using the MOE package (www.chemcomp.com). To match experimental conditions, the sequence of the structure was reverted to that of a rat (Uniprot code: P29067).

To simulate rebinding of the distal C-tail (Fig 1D), we have modeled the βarr2 C-tail starting from residue 395 to residue 397 and then manually displaced the C-terminal part of the tail from the core of βarr2. Afterward, the system was simulated in 35 replicates for 200 ns, allowing the C-tail to spontaneously assume an inactive conformation (positional constraints applied to backbone atoms of βarr2 residues until residue 388, allowing C-tail flexibility). The frames were clustered using the internal algorithm in VMD [75] based on the RMSD of the Arg394 heavy atoms (cutoff 3.7).

The intermediate βarr2-C7pp2 complex was generated by docking the peptide into a representative conformation extracted from the main cluster (Fig 1D). The peptide (residues 348-359) was docked using the HADDOCK 2 web server [76]. The binding site was defined using positions within βarr2 canonically associated with GPCR C-tail binding. The final pose was selected based on scoring as well as guidelines from available βarr-C-tail complexes. Subsequently, the whole βarr2 C-tail was modeled, and the system was allowed to relax (35 × 300 ns, positional constraints applied to backbone atoms of βarr2 residues until residue 388 as well as the docked peptide backbone). Aiming to analyze only frames where the system was equilibrated, the last 50 ns of each run was taken into account. The resulting frames were clustered based on the RMSD of the Arg394 heavy atoms (cutoff 1). From these frames, we extracted a βarr2-C7pp2 conformation for subsequent simulation runs. When studying the stability of the βarr2-C7pp2 complex, as well as the impact of the E358A mutant and lack of the C7pp2 tail (Fig 3B) the systems were simulated in five replicates for 300 ns.

All generated systems were solvated (using TIP3P model water) and neutralized with NaCl ions (0.15 M concentration) using the CHARMM-GUI web server [77]. System parameters were obtained from the Charmm36M forcefield [78,79]. Simulations were carried out using Acemd3 [80]. Each system initially underwent 20 ns of equilibration in conditions of constant pressure (NPT) with positional constraints applied to the backbone atoms. For production runs, the systems were simulated in conditions of constant volume (NVT) using a 4 fs timestep. Simulations were analyzed using VMD [75]. Simulations are made available at the online GPCRmd resource [81]: (https://www.gpcrmd.org/dynadb/publications/1525/)

### Cloning, protein expression, and purification

CXCR7 phosphopeptides (C7pp2 and C7pp3) for clathrin binding assay and ITC experiments were synthesized from NovoPep and Tufts University Core Facility (Fig 1A). The *Rattus norvegicus* wild-type βarr2 was cloned into the pET-28a expression vector and then were transformed into *Escherichia coli* BL21(DE3)pLysS cells (Invitrogen). Cells harboring the plasmids were grown in Luria Bertani (LB) broth containing 70 μg mL$^{-1}$ chloramphenicol and 50 μg mL$^{-1}$ kanamycin until

the optical density (at 600 nm) reached 1.0–1.2 at 37 °C. Further, protein expression in the cells was induced by using 0.1 mM isopropyl-β-D-1-thiogalactopyranoside, after which the cells were incubated for 16 h at 16 °C.

To isolate the βarr2 protein fused to an N-terminal His$_6$-tag, cells were collected by centrifugation at 4,000$g$ at 4 °C for 10 min and resuspended in buffer A (20 mM Tris-HCl pH 8.0, 500 mM NaCl, and 5 mM imidazole) supplemented with 1 mM phenylmethanesulfonylfluoride. The cells were lysed by passing through a Microfluidizer (Microfluidics, Westwood, MA, USA), and the cell lysate was cleared by centrifugation. The supernatant was collected and loaded onto a Ni-sepharose affinity column (GE Healthcare, Little Chalfont, UK) and washed with buffer A. The protein was eluted using buffer A supplemented with a gradient of imidazole concentrations ranging from 100 mM to 1 M. The eluted fractions were desalted into buffer B (20 mM Tris-HCl pH 8.0 and 5 mM β-mercaptoethanol) containing 100 mM NaCl. Subsequently, further purification was carried out using affinity chromatography with a HiTrap heparin column (GE Healthcare). The proteins were eluted from the heparin column using buffer B supplemented with 1 M NaCl. To achieve further purification, eluted βarr2 was concentrated and subjected to gel filtration using a HiLoad 16/60 Superdex 200 prep-grade column (GE Healthcare). The column was pre-equilibrated with buffer B containing 200 mM NaCl and protein-containing fractions were collected. Protein was concentrated to approximately 12 mg mL$^{-1}$. The protein concentration was determined by measuring the absorbance at 280 nm.

### Clathrin binding assay

To evaluate the binding of clathrin to wild-type βarr2, a clathrin binding assay was conducted in the absence or presence of a 5:1 (peptide:βarr2) molar ratio of ligand (V$_2$Rpp, C7pp2, and C7pp3). For each condition, 20 µg of βarr2 was incubated with the respective peptide in a binding buffer consisting of 20 mM Tris-HCl pH 8.0, 100 mM NaCl, and 1 mM DTT, at 4 °C for 1 h. Following the incubation, the volume of each mixture was adjusted to 100 µL using the binding buffer. Subsequently, 30 µL of GST beads, containing 30 µg of GST-clathrin, were added to the mixture, which was then agitated for 2 h at 4 °C. The GST beads with bound GST-clathrin were centrifuged at 20,000$g$. The beads were subsequently washed with 1 mL of the binding buffer five times. Following the washing steps, the beads were incubated with 50 µL of 25 mM GSH buffer. Clathrin binding to βarr2 was measured by Western blot analysis with an anti-his-tag mouse mAb (1:5000, MBL). Secondary antibodies used were anti-mouse IgG, HRP-linked Antibody (Cell Signaling Technology). Blots were re-probed with a GST mouse mAb (1:5000, Cell Signaling Technology) to ensure equal loading of GST-clathrin for each reaction.

### Isothermal titration calorimetry (ITC)

ITC experiments were performed using Affinity ITC instruments (TA Instruments, New Castle, DE, USA) at 298 K. 200 µM of wild-type βarr2, which was prepared in a buffer containing 20 mM HEPES pH 7.0 and 200 mM NaCl was degassed at 298 K before measurements. Using a micro-syringe, 2.5 µL of 1 mM C7pp2 peptide solutions was added at intervals of 200 s to the wild-type βarr2 solution, in the cell with gentle stirring. The ITC experiments for the βarr2 mutants (R166A and R394A) and the C7pp2 mutants (C7pp2-1, C7pp2-2, and C7pp2$^{E358A}$) (S6 Fig) were conducted under the same running conditions.

### Surface expression assay

To study the receptor surface expression of CXCR7$^{WT}$ and CXCR7$^{E358A}$, a whole cell-based receptor surface ELISA was performed as previously discussed [82]. Briefly, cells transiently transfected with either CXCR7$^{WT}$ or CXCR7$^{E358A}$ were seeded in a 24-well plate (corning) pre-coated with 0.01% poly-D-lysine at a density of 0.2 million cells per well and incubated at 37 °C for 24 h. After 24 h, the plate was taken out and the growth media was removed. The cells were once washed with ice-cold TBS followed by fixation with 4% Paraformaldehyde (w/v in TBS) on ice for 20 min. Thereafter, cells were washed three times (400 µL in each wash) with TBS followed by blocking with 1% BSA prepared in TBS at room

temperature for 90 min. Subsequently, the cells were incubated with anti-FLAG M2-HRP (Sigma) antibody for 90 min. Antibody was prepared in 1% BSA in TBS at a dilution of 1:5,000. Following antibody incubation, cells were washed three times with 1% BSA (in TBS). Thereafter, cells were incubated with 200 µL of TMB-ELISA (Thermo Scientific) till the light blue color appeared and the signal was quenched by transferring 100 µL of colored solution to another 96-well plate containing 100 µL of 1 M $H_2SO_4$. Absorbance was measured at 450 nm using a multi-mode plate reader. For normalization of signal intensity, cell density was estimated by using a mitochondrial stain Janus green B. Afterward, TMB was removed and cells were washed two times with 200 µL of TBS. After washing, the cells were incubated for 15 min with 0.2% (w/v) Janus Green (Sigma). Excess stain was removed by washing with distilled water (three times). The stain was eluted by adding 800 µL of 0.5 N HCl per well. 200 µL of eluted solution was transferred to a 96-well plate and absorbance was recorded at 595 nm. The signal intensity was normalized by calculating the ratio of A450/A595 values. For data normalization, ratio of A450/A595 value of pcDNA transfected cells reading was considered as 1 and calculated receptor expression with respect to pcDNA. Data were analyzed by using column in GraphPad Prism software.

## Confocal microscopy

For visualizing βarr2 recruitment and its trafficking, HEK293 cells were transfected with 5 µg of CXCR7$^{WT}$ or CXCR7$^{E358A}$ along with 2 µg of βarr2-mYFP mammalian constructs using a polyethylenimine reagent (21 µL) in 10 cm plates. After 6 h, the transfected HEK293 cells in FBS-deficient DMEM media were now removed and replaced with 10% FBS-supplemented DMEM media (Gibco). Post 24 h, transfected cells were trypsinized and seeded onto poly-D-lysine (Sigma) precoated glass bottom confocal dishes (GenetiX) at a density of 1 million per plate. Simultaneously, with the same transfected cells, another 24-well plate was seeded at 0.1 million per well for comparing the surface expression of CXCR7$^{WT}$ and CXCR7$^{E358A}$. After overnight culturing the cells in confocal dishes, cells were now starved in FBS-deficient DMEM media for 4 h. Cells were visualized under the microscope and stimulated with CXCL12 (100 nM, Peprotech). Live cell microscopy was done on the cells, and images were captured at multiple time points. The instrument used was Zeiss LSM 710 NLO confocal microscope where cells were housed on a temperature and $CO_2$-controlled platform with a motorized XY stage. A diode laser at 488 nm laser line was used for exciting mYFP, and an emitted signal was detected with a 32×array GaAsP descanned detector (Zeiss). For all experiments, microscopic settings, including laser intensity and pinhole slit were kept in the same range.

## Tango assay

Tango assay measures βarr2 recruitment to the receptor. HTLA cell line, a derivative of HEK293 cell line stably expressing a tTA-dependent luciferase reporter gene and βarr2-TEV fusion gene was maintained in Dulbecco's Minimum Essential Media supplemented with 10% FBS, 2 µg mL$^{-1}$ puromycin and 100 µg ml$^{-1}$ hygromycin B at 37 °C in 5% $CO_2$. Briefly, HTLA cells at a density of $3 \times 10^6$ cells per 10 cm plate were transfected with 7 µg of CXCR7$^{WT}$ or CXCR7$^{E358A}$ constructs. After 24 h of transfection, cells were trypsinized and seeded at a density of 0.05 million cells per well in a white 96-well polystyrene micro-plate. After another 16 h, cells were stimulated with varying doses of CXCL12 (0.32 µM to 0.1 pM). The ligand concentrations were prepared in incomplete media devoid of FBS. After incubation with ligand, media was aspirated and 100 µL of luciferin (0.5 mg mL$^{-1}$ in HBSS buffer) was added to each well and the plate was read for luminescence. The data was normalized with respect to the highest dose of ligand in CXCR7$^{WT}$ after basal correction, and then analyzed using nonlinear regression in GraphPad Prism software.

## HDX-MS analysis

Wild-type βarr2 was prepared at a final concentration of 100 µM in a buffer consisting of 20 mM HEPES pH 7.4 and 150 mM NaCl. For peptide binding, 500 µM of C7pp2 was added to βarr2 and incubated at room temperature for 1 h. To initiate hydrogen/deuterium exchange, 2 µL of protein samples were mixed with 28 µL of D$_2$O buffer (20 mM HEPES

pH 7.4, 150 mM NaCl, and 10% glycerol in $D_2O$) and incubated on ice for 10, 100, 1000, or 10 000 s. At the indicated time points, the reaction was slowed down by adding 30 µL of ice-cold quench buffer (100 mM $NaH_2PO_4$ pH 2.01). Non-deuterated samples were prepared by mixing 2 µL of the protein sample with 28 µL of $H_2O$ buffer (20 mM HEPES pH 7.4 and 150 mM NaCl in $H_2O$) and subsequently quenching the reaction with 30 µL of ice-cold quench buffer. The quenched samples were subjected to online digestion by passing them through an immobilized pepsin column (2.1 mm × 30 mm) at a flow rate of 100 mL min⁻¹ using 0.05% formic acid in $H_2O$ as the mobile phase at 12 °C. After the digestion process, the peptide fragments were collected on a C18 VanGuard trap column (1.7 mm × 30 mm) for desalting with 0.05% formic acid in $H_2O$. The proteins were subsequently separated using ultra-pressure liquid chromatography on an ACQUITY UPLC C18 column (1.7 µm, 1.0 mm × 100 mm) at a flow rate of 40 mL min⁻¹. The separation was achieved by employing an acetonitrile gradient generated by two pumps, initially starting with 8% B and gradually increasing to 85% B over a duration of 8.5 min. Mobile phase A consisted of 0.15% formic acid in $H_2O$ and mobile phase B comprised 0.15% formic acid in acetonitrile. To minimize the back exchange of deuterium to hydrogen, all components involved in the analysis, including the sample, solvents, trap, and UPLC column, were maintained at a pH of 2.5 and 0.5 °C. Mass spectral analyses were performed with a Xevo G2 QTof equipped with a standard ESI source (Waters, Milford, MA, USA). The mass spectra were acquired in the positive ion mode in the range of $m/z$ 100–2000 for 12 min. Peptides were identified in non-deuterated samples with ProteinLynx Global Server (PLGS) 2.4 (Waters, Milford, MA, USA). The following parameters were applied: monoisotopic mass, non-specific for the enzyme while allowing up to 1 missed cleavage, MS/MS ion searches, automatic fragment mass tolerance, and automatic peptide mass tolerance. The searches were conducted with the variable methionine oxidation modification and the peptides were filtered with a peptide score of 6. To analyze the HDX-MS data, the deuterium content in each peptide was determined by measuring the centroid of the isotopic distribution using DynamX 2.0 (Waters, Milford, MA, USA). All measurements were performed in three independent experiments, and statistical significance was assessed using one-way ANOVA. Back-exchange levels were not corrected because the analyses compared different states. The details of HDX-MS data are described in S1 Data.

## Supporting information

**S1 Fig. Spontaneous inactivation of the C-tail observed during simulations of βarr2 with a manually displaced C-tail.** The distance between R394 and D299 (forming polar interactions in the inactive state of βarr2) monitored within MD simulations (35 × 200 ns). The data underlying the graphs shown in the figure can be found in S1 Data.
(TIF)

**S2 Fig. Impact of the number of replicates on the shape and population of the R394 main cluster. (A, B)** Main cluster conformations of R394 during spontaneous inactivation (A) and Docking pose optimization (B) are shown in blue sticks. The population of each R394 cluster is indicated.
(TIF)

**S3 Fig. The distance between R394 and D299 (forming polar interactions in the inactive βarr2 state) monitored within MD simulations (5 × 300 ns).** The data underlying the graphs shown in the figure can be found in S1 Data.
(TIF)

**S4 Fig. Structural flexibility explored by the βarr2 C-tail (blue) and C7pp2 (red) explored in MD simulations (5 × 300 ns, 1 snapshot every 10 ns).**
(TIF)

**S5 Fig. Stability of polar interactions between C7pp2 and βarr2. (A, B)** The distance between residues in site I (pS350 and pT352 of C7pp2; R77, R148, and R166 of βarr2) and site II (E358 of C7pp2 and R394 of βarr2) are monitored

during the whole simulation run (panel A, 5 × 300 ns) and the last 50 ns of each simulation replicate (panel B, 5 × 50 ns). For site I, the distance between the phosphorus (P) atom of the phosphorylated residues (pS350 and pT352) and the carbon atom in the guanidinium group (Cζ) of the corresponding arginine residues (R77, R148, and R166) are plotted. For site II, the distance between the Cδ atom of E358 and the Cζ atom of R394 is plotted. The data underlying the graphs shown in the figure can be found in S1 Data.
(TIF)

**S6 Fig. The two binding interfaces of βarr2IM-C7pp2, including site I and site II.** ITC experiments showing the effect of mutations in the binding interfaces between βarr2 and C7pp2. The detailed sequences of C7pp2 mutants used for the ITC assay are shown in the lower right panel. Purified βarr2 was incubated with increasing peptide concentrations, and the binding parameters were calculated based on the dose-response curve. The data underlying the graphs shown in the figure can be found in S1 Data.
(TIF)

**S7 Fig. Comparison of HDX-MS data of βarr2 alone and βarr2-C7pp2. (A, B)** The deuterium uptake profiles were mapped onto the structure of βarr2 (panel A) and βarr2-C7pp2 (panel B). Deuterium incorporation after 10 and 10,000 seconds of $D_2O$ buffer incubation is indicated by a color code. The color legend represents the level of deuterium uptake, and uncovered regions are shown in gray. N-terminal part of βXX region in the structure is highlighted as black dotted circle.
(TIF)

**S1 Movie. Molecular dynamics simulation of the spontaneous inactivation of the βarr2 C-tail.**
(MP4)

**S1 Data. Data underlying figures.**
(XLSX)

**S1 Raw Images. Unprocessed original images for Fig 1B.**
(PDF)

## Ackonwledgments

We thank Dr. Mithu Baidya and Parishmita Sarma from IIT Kanpur for the Tango assay.

## Author contributions

**Conceptualization:** Jeong Seok Ji, Yaejin Yun, Jana Selent, Hyung Ho Lee.

**Data curation:** Tomasz Maciej Stepniewski, Miguel Dieguez Eceolaza.

**Formal analysis:** Miguel Dieguez Eceolaza.

**Funding acquisition:** Ka Young Chung, Jana Selent, Hyung Ho Lee.

**Investigation:** Jeong Seok Ji, Yaejin Yun, Tomasz Maciej Stepniewski, Hye-Jin Yoon, Kyungjin Min, Ji Young Park, Chiwoon Chung, Ka Young Chung, Jana Selent, Hyung Ho Lee.

**Project administration:** Hyung Ho Lee.

**Supervision:** Jana Selent, Hyung Ho Lee.

**Writing – original draft:** Jeong Seok Ji, Yaejin Yun, Tomasz Maciej Stepniewski, Jana Selent, Hyung Ho Lee.

**Writing – review & editing:** Jeong Seok Ji, Yaejin Yun, Tomasz Maciej Stepniewski, Hye-Jin Yoon, Kyungjin Min, Ji Young Park, Chiwoon Chung, Ka Young Chung, Jana Selent, Hyung Ho Lee.

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
