## [Editor Report · Decision Letter 0]

5 Nov 2024

Dear Dr Lee,

Thank you for submitting your manuscript entitled "An arginine switch drives the stepwise activation of β-arrestin" for consideration as a Research Article by PLOS Biology.

Your manuscript has now been evaluated by the PLOS Biology editorial staff as well as by an academic editor with relevant expertise and I am writing to let you know that we would like to send your submission out for external peer review.

Once your full submission is complete, your paper will undergo a series of checks in preparation for peer review. After your manuscript has passed the checks it will be sent out for review. To provide the metadata for your submission, please Login to Editorial Manager (https://www.editorialmanager.com/pbiology) within two working days, i.e. by Nov 07 2024 11:59PM.

Kind regards,

Suzanne

Suzanne De Bruijn, PhD

Associate Editor

PLOS Biology

sbruijn@plos.org

---

## [Decision Letter · Decision Letter 1]

16 Jan 2025

Dear Dr Lee,

Thank you for your patience while your manuscript entitled "An arginine switch drives the stepwise activation of β-arrestin" was peer-reviewed at PLOS Biology. Please also accept my sincere apologies for the long delay in sending you our decision. The manuscript has now been evaluated by the PLOS Biology editors, an Academic Editor with relevant expertise, and by three independent reviewers.

The reviews are attached below. As you will see, the reviewers find the conclusions potentially interesting, however they also raise several concerns regarding the evidence provided to support them. Reviewer 1 mentions that you should include a domain layout of barr2 with the domain and structural features discussed, and suggests also rearrangements for some of the figures. Reviewer 2 thinks that the boxplot presented is inconsistent as the conditions don’t have replicative plots and that it is not confirmed that the differences are significant. In addition, this reviewer mentions that unless further analysis of other receptor sequences and potential simulations are performed, the conclusions cannot be generalised to other GPCRs, and that there is no evidence showing that the ACKR3 and arrestin interaction is specific. Reviewer 3 raises concerning issues with the MD simulations and thinks you should provide further evidence about the convergence of the simulations. In addition, this reviewer mentions that the ITC experiments are not properly described, the Tango assay doesn’t show that the intermediate state exists and that the observed differences in the HDX experiments are too small and only observed for the first time point.

In light of the reviews and after discussing them with the Academic Editor at length, we have decided to give you a chance at addressing the concerning critiques raised by the reviewers and invite you to revise the work to thoroughly address the reviewers' reports. Please note that all the concerns should be satisfied in full and the conclusions supported in order for us to consider the manuscript further for publication. Given the extent of revision needed, we cannot make a decision about publication until we have seen the revised manuscript and your response to the reviewers' comments. Your revised manuscript is likely to be sent for further evaluation by all or a subset of the reviewers.

**IMPORTANT - SUBMITTING YOUR REVISION**

3. Resubmission Checklist

a) *PLOS Data Policy*

b) *Published Peer Review*

d) *Blurb*

Please also provide a blurb which (if accepted) will be included in our weekly and monthly Electronic Table of Contents, sent out to readers of PLOS Biology, and may be used to promote your article in social media. The blurb should be about 30-40 words long and is subject to editorial changes. It should, without exaggeration, entice people to read your manuscript. It should not be redundant with the title and should not contain acronyms or abbreviations. For examples, view our author guidelines: https://journals.plos.org/plosbiology/s/revising-your-manuscript#loc-blurb

Sincerely,

Ines

--

Ines Alvarez-Garcia, PhD

Senior Editor

PLOS Biology

Reviewers' comments

Rev. 1: Jennifer Cash – note that this reviewer has signed her review.

This manuscript describes an intermediate state in the activation of barr2 through binding the phosphorylated C-terminal tail of a GPCR using molecular dynamics simulations, ITC, a cell-based assay, and HDX-MS. Specifically, the authors are interested in examining how the sequestered barr2 C-terminal tail, which binds the N-domain and stabilizes the inactive state, is released upon interaction with GPCRs. Previous structural data has been limited to the active and inactive states of barr, leaving the mechanisms occurring in the intermediate states unknown. Using MD, the authors find that a peptide representing a phosphorylated GPCR C-terminal tail binds to barr2 while the barr2 C-tail is still bound to the N-domain in an intermediate state, identifying a previously unrecognized binding mode. The manuscript is well-written and appears overall technically sound. I have the following suggestions to improve the presentation and interpretability of the data:

* I recommend including a domain layout of barr2 with the domain and structural features discussed in the manuscript highlighted.

* Line 106-107, how do the authors know that C7pp2 is binding and inducing an intermediate state? This seems like a jump in rationale.

* In figure 1C, I would recommend not coloring the aspartate sidechains red, or coloring the oxygens a different shade of red.

* In the legend for figure 3, please include information on the statistical analysis that was performed and significance.

* I suggest moving figure S5 into the main text. I find it to be much more informative and clear with regards to following the text than current figure 3A. If 3A was replaced and 3B made smaller, I think S5 would fit within 3.

* In figure 3C, what does "surface" and "endosomal" refer to? How are these treatments different? This information should be included either in the figure legend or the results section. Some description of the assays used in figure 3C-E should be included in the results section (lines 212-216) to help the reader understand why these assays were used.

* The results section describing the HDX-MS data needs further explanation. Why could data not be obtained on the distal C-tail region? This could signify that something is wrong with the sample, unless further information indicating otherwise is provided. Is the 382-389 region the only part of this tail that data could be obtained on? I would recommend that the authors also show the complete HDX-MS data showing the overall dynamics of barr2 and the peptide plotted onto each structure (alone and in complex, either one chosen time point or an average), for completeness and show this in the supplemental. Per peptide deuterium uptake levels are shown in an Excel file in the supplemental, but it is not easy for the reader to interpret the overall dynamics of barr2 by looking at this. Also, what is the relevance of the decreased exchange rates in the middle loop, lariat loop, and back loop? These features are not discussed elsewhere in the text. Please include a discussion of this.

* I like the mechanistic model in figure 5 - well done!

Rev. 2:

In the manuscript entitled "An arginine switch drives the stepwise activation of β-arrestin" by Jeong Seok Ji, et al. the authors use molecular dynamics, ITC binding assays, fluorescence microscopy, and HDX experiments to describe an interaction between β-arrestin and a phosphopeptide of ACKR3 that they claim is important for the activation of β-arrestins. The paper does a good job of characterizing this interaction and providing multiple sources to support the importance of a conserved arginine residue (R394) and its role in destabilizing the polar core and subsequent releasing of the C-terminal tail to allow for β-arrestin activation. While the authors describe a mechanism for the binding of the ACKR3 C-terminal peptide to β-arrestin2 (namely a glutamate residue in the C-terminus of ACKR3 that they propose interacts with R394 of β-arrestin2) they fall short in relating this broadly to GPCRs at large. While the introduction and discussion sections are framed in terms of general mechanisms for arrestin activation by GPCRs, no analysis is done to assess the presence of comparable residues or motifs in other GPCR C-terminal sequences. This omission must be addressed to justify any discussion of general GPCR/arrestin activation mechanisms.

Major comments

* What is V2Rpp? It is never described, and its relevance to the ACKR3 peptides that are the focus of the study create confusion at the outset that is never resolved.

* Page 11, Lines 33-35 "Our study uncovers a previously unknown molecular switch that, upon engagement, destabilizes the polar core as a crucial step in the GPCR-induced βarr activation.": While the findings in this paper do expand on the importance of a conserved arginine residue in β-arrestins, I would hardly say this was a previously unknown molecular switch. The authors themselves stated "Our analysis specifically focused on Arg394 within the distal C-tail, as previous experimental findings have underscored the conformational rearrangement of this residue as an important step in C-tail displacement50 ".

* Figure 3B - The data shown on the boxplot is inconsistent. All conditions should have replicate points shown as with the first condition (red box). Additionally, some significance analysis could be done between these conditions to convince the reader that these differences are meaningful as the replicates from the WT conditions seem to span the entire range, albeit slightly skewed to the upper distances.

* Page 22, Line 290 "While our experiments elucidate the activation mechanism of βarr2 by the receptor Rp-tail": The paper describes a potential activation mechanism for the ACKR3 C-terminal peptide and β-arrestin2. This does not necessarily apply broadly to other GPCRs and further analysis of other receptor sequences and potential simulations would have to be done to make this claim. ACKR3 is intrinsically biased toward arrestin, and the authors offer no evidence that this interaction isn't unique to ACKR3. I don't particularly think this is unique to ACKR3 as only the C-terminus was used and it's not particularly different from other GPCRs, but the authors don't even mention this potential pitfall, and this leaves the central claim with some doubt.

Minor Comments

* Page 11, Lines 29-30 "we reveal how specific interactions with a prototypical GPCR": ACKR3 is not a "prototypical" GPCR. It doesn't couple to G-proteins and is intrinsically arrestin biased. The A stands for atypical.

* Page 12, Lines 44-46 "The removal of GPCRs from the cell surface through βarr-mediated internalization prevents them from transmitting extracellular signals into cells.": This sentence is followed by a contradictory statement and should be revised to clarify and harmonize the intended meaning (e.g. "continued transmission of extracellular")

* Page 16, Lines 151-154 "Notably, the structural model is supported by isothermal titration calorimetry (ITC) experiments (Fig. 2B). In ITC, βarr2 exhibits a biphasic binding pattern with C7pp2, and data fitting to a two-site binding model indicates that C7pp2 contains two distinct binding motifs for βarr2.": Fitting values (KD values) are not shown in the text or in Figure 2B and should be added. KD values are shown in the supplement for other ITC conditions.

* Figure 3A - Side chains should match the corresponding ribbon colors as in Figure 2.

Rev. 3:

This manuscript by Ji and coworkers describes a combination of molecular dynamics simulations (MD), biochemical and cell-based experiments to elucidate the activation mechanism of β-arrestin.

Arrestin activation occurs by the binding of the phosphorylated tail or intracellular loops of G protein-coupled receptors (GPCRs). The structures of β-arrestins (also called arrestin2 and arrestin3) in complex with phosphorylated GPCRs or respective phosphopeptides have been elucidated and show that the phosphopeptide forms an intramolecular beta-sheet with arrestin strand beta-1. The peptide thereby replaces arrestin beta-strand 20, which in (inactive) apo arrestin forms an intramolecular beta-sheet with strand beta-1. As a consequence, beta-20 and the entire C-terminus are released from the arrestin core, the arrestin N- and C-domains rotate slightly against each other and the arrestin loop conformations change.

While this activated state is known, it is unclear whether the binding of the peptide occurs via intermediate steps. To elucidate such intermediate steps, the authors use mostly a shortened version of a phosphopeptide derived from the carboxyl terminus of CXCR7, C7pp2, to study the interactions with βarr2. C7pp lacks the last three residues STK of the C-tail, two of which (ST) can also be phosphorylated. A second peptide C7pp3 comprises these residues. The authors carry out MD simulations as well as biochemical and cellular experiments, from which they claim that the C7pp2 peptide displaces as a first step R394 while still keeping the beta-20.beta-1 sheet intact. This would then be an intermediate step during arrestin activation.

However, as indicated below, this evidence is not solid and the reasoning is flawed. For this reason, I cannot recommend this manuscript for publication.

In detail:

1. MD simulations: the MD simulations were carried out for sets of 7 x 300 ns or 5 x 200 ns. This is much too short to follow a beta-strand release or replacement from a beta-sheet. Typical opening times of individual beta-hairpins are on the order of at least 10 microseconds. The authors give no evidence about the convergence of their simulations. Thus, it is unclear what to conclude much from any of the MD results.

2. ITC experiments: Fig. 2B shows a biphasic ITC profile. The horizontal axis is not labeled. I assume it is the molar ratio. No stoichiometry is indicated. The biphasic behavior is used as evidence for a two-site (step-wise) binding mechanism. However, this does not make sense. If the horizontal axis is indeed the molar ratio, then the reaction would give two C7pp2 peptides bound to one βarr2 molecule. How can this be? A GPCR has only one tail. The experiment also lacks a description how the fit curve was obtained. Such a biphasic fit must involve a 2:1 chemical reaction.

3. Tango assay (Fig. 3 C-E): the assay shows that CXCR7-E354 is important for βarr2 recruitment, but not more. I.e. it does not give evidence that the claimed intermediate state exists.

4. HDX experiments (Fig. 4): The observed differences between C7pp2-bound and -free βarr2 are in general very small. The claimed significant differences for decreased exchange (blue asterisks) are particularly small and are observed only for the first time point, which may be an experimental variation after mixing. More importantly as indicated in the methods 2 μL of protein samples were mixed with 28 μL of D2O buffer. So, the final deuterium uptake should be 1-2/28 = 93%. However, the uptake is maximally 60% and varies strongly between the detected stretches albeit saturation seems to be achieved. This is completely unclear. As a small additional point, it is also unclear what the following sentence in the figure legend means 'Differences smaller than 0.2 Da were not considered significant.'

---

## [Decision Letter · Decision Letter 2]

22 May 2025

Dear Dr Lee,

Thank you for your patience while we considered your revised manuscript "An arginine switch drives the stepwise activation of β-arrestin by CXCR7" for publication as a Research Article at PLOS Biology. This revised version of your manuscript has been evaluated by the PLOS Biology editors, the Academic Editor and one of the original reviewers. The Academic Editor also checked your responses to the other reviewers.

Based on the review and on our Academic Editor's assessment of your revision, we are likely to accept this manuscript for publication, provided you satisfactorily address the data and other policy-related requests stated below my signature.

We expect to receive your revised manuscript within two weeks.

*Published Peer Review History*

*Press*

Sincerely,

Ines

--

Ines Alvarez-Garcia, PhD

Senior Editor

PLOS Biology

Fig. 2B, C; Fig. 3B, D, E; Fig. 4; Fig. S1; Fig. S3; Fig. S5A, B and Fig. S6

CODE POLICY

We require the original, uncropped and minimally adjusted images supporting all blot and gel results reported in an article's figures or Supporting Information files. We will require these files before a manuscript can be accepted so please prepare and upload them now - we require the original gels for Fig. 1B. Please carefully read our guidelines for how to prepare and upload this data: https://journals.plos.org/plosbiology/s/figures#loc-blot-and-gel-reporting-requirements

Reviewers' comments

Rev. 2:

The authors provide thorough and extensive responses to the comments raised by the reviewers which significantly improve the clarity of the manuscript. No additional comments or concerns.

---

## [Editor Report · Decision Letter 3]

11 Jul 2025

Dear Dr Lee,

Thank you for the submission of your revised Research Article entitled "An arginine switch drives the stepwise activation of β-arrestin by CXCR7" for publication in PLOS Biology. On behalf of my colleagues and the Academic Editor, Carole Parent, I am delighted to let you know that we can in principle accept your manuscript for publication, provided you address any remaining formatting and reporting issues. These will be detailed in an email you should receive within 2-3 business days from our colleagues in the journal operations team; no action is required from you until then. Please note that we will not be able to formally accept your manuscript and schedule it for publication until you have completed any requested changes.

PRESS

Sincerely, 

Ines

--

Ines Alvarez-Garcia, PhD

Senior Editor

PLOS Biology
